# Characteristics and risk factors for sibling incest

**Kelly M. Babchishin**[1], **Emma J. Holmes**[1], **Rainer Banse**[2]*, **Lisa Huppertz**[2], **Michael C. Seto**[1,3]

**1** Department of Psychology, Carleton University, Ottawa, Ontario, Canada, **2** Department of Psychology, Universität Bonn, Bonn, North Rhine-Westphalia, Germany, **3** Royal Ottawa Health Care Group, Ottawa, Ontario, Canada

☯ These authors contributed equally to this work.

* banse@uni-bonn.de

**Data Availability Statement:** The data underlying the results presented in the study are available from https://osf.io/6qmyk/?view_only= 3575930005f94fee9b9fc624c6087b4f.

## Abstract

Sibling sexual behaviour, despite historical and cross-cultural incest taboos and biologically driven incest avoidance, poses a persistent problem. We tested factors theorized to be associated with sibling incest in a cross-sectional online survey of 1,863 respondents with siblings mainly from North America and Germany. We found that 13% of participants reported engaging in sexual contact with a sibling, typically starting at the age of 10, and that step-siblings and half-siblings were more likely to engage in sibling incest than full siblings. Curiosity and games were the primary motivators; being coerced was more prevalent among female and younger participants. The study underscores both individual (e.g., impulsivity, concurrent childhood sexual behaviour problems) and family level factors (e.g., presence of step-sibling, positive attitudes toward nudity, sexual abuse by parent) influencing liability to engage in sexual behaviours with a sibling. Findings were robust across English- and German-speaking participants, suggesting our results are generalizable. Professionals addressing problematic child sexual behaviour should assess for concurrent sibling incest, and evaluate positive family attitudes towards nudity, sexual abuse by parents, and reduced disgust to sibling incest as potential risk factors for sibling incest. The findings stress the need for comprehensive sexual education in blended households, where age gaps and diminished genetic relatedness contribute to sibling sexual behaviour.

## Introduction

It is an evolutionary and social puzzle why some individuals engage in sex with their siblings despite widespread incest taboos (religious or cultural norms about acceptable sexual relationships) and biologically driven incest avoidance (sexual indifference or aversion among close relatives). Solving this puzzle is critical because, every year, millions of intrafamilial child sexual abuse cases result in legal or social service actions, with siblings being more common perpetrators than fathers or stepfathers [1, 2]. Indeed, most sexual offences committed by youth are committed against younger, related children, particularly younger siblings [3]. The

**Funding:** This research was supported by the Canadian Institutes of Health Research (CIHR) Banting fellowship (K. Babchishin) and the University of Ottawa Medical Research Fund (201605; Babchishin & Seto). E. Holmes is supported in part by funding from the Social Sciences and Humanities Research Council and Ontario Graduate Scholarship. Any opinions, findings, and conclusions or recommendations expressed in this material are those of the authors and do not necessarily reflect the views of the funding bodies. The funders had no role in study design, data collection and analysis, decision to publish, or preparation of the manuscript. The other authors received no additional funding.

**Competing interests:** The authors have declared that no competing interests exist.

sequelae of sibling incest can be severe and wide-ranging, including mental health problems, substance use, sexual dysfunction, and risky behaviour, including risky sexual behaviour [1].

Yet, we know remarkably little about characteristics and risk factors for sibling incest [4, 5]. Seto [6] has identified factors that could explain sibling incest in a review of existing theories. Many of these theories focus on cues of relatedness as triggers for incest avoidance, including Westermarck's [7] hypothesis regarding the effects of close proximity in early childhood, Lieberman et al.'s hypothesis about maternal-neonatal association (observing younger siblings as babies with one's mother) [8], and Lieberman et al.'s [9] hypothesis about physical resemblance being a cue of relatedness. Of these, Westermarck's [7] close proximity hypothesis has received the most empirical support, though Rantala and Marcinkowska [10] have argued that close proximity is not sufficient because multiple kinship cues inform incest avoidance. These distal explanations of sibling incest (e.g., focused on factors present in childhood) have been hypothesized to manifest through the proximal mechanisms of disgust at the idea of sex with a sibling and a lack of sexual attraction to siblings.

More clinical theories of incest focus on more proximal factors, suggesting that family dysfunction, antisocial tendencies, atypical sexuality, or social unpopularity can explain sibling incest [6, 11]. Some of these factors have empirical support: for example, Griffee et al. [12] surveyed a student sample of 1,821 women and 1,064 men recruited from US colleges and found that sibling incest was more likely when siblings shared a bed or took baths together, family nudity was common, or parent-child incest had taken place. Approximately a quarter of sibling incest (38/137 participants) was characterized as coercive by participants, but the nature of this coerciveness was not explored [12].

Sexual behaviours between siblings can also be part of normal sexual development, but what entails normative (common) child sexual behaviours is under-researched [13]. Normative child sexual behaviour is often described as exploratory play with children of similar ages and includes non-intrusive sexual behaviours [14]. Friedrich et al. [15] *Child Sexual Behavior Inventory*, outlines sexual behaviours that are not normative in childhood, and includes behaviours such as passionate kissing (open-mouthed kissing involving tongue contact) as well as anal, vaginal, and oral penetrative sexual behaviours. Factors that may be relevant to non-normative sibling sexual behaviour include a larger age difference between siblings, the presence of coercion, and whether vaginal or anal penetration took place. Longitudinal research has identified factors associated with the perpetration of sexual offences against children [16]. In a meta-analysis of 29,450 sexual offending individuals sampled across 82 studies, atypical sexual interests (e.g., sexual interest in children) and antisocial tendencies were the two strongest predictors of sexual reoffending. Being male is also a risk factor for perpetrating sexual offences [17], including incest offences [18].

There are no rigorous studies examining the prevalence of and correlates for sibling incest on a non-clinical, non-university, or non-forensic group of individuals with siblings. Available studies do not distinguish between sibling types (e.g., full vs. step-siblings), yet the composition of families has changed drastically in recent years. In Canada, for example, it is now estimated that 1 out of 15 children will be living with at least one step-sibling or half-sibling [19]. As such, the primary aim of the present research was to examine the prevalence and risk factors of sibling incest (i.e., sexual behaviour between siblings, including passionate kissing, touching sexual organs, masturbation, receiving or giving oral sex, vaginal or anal intercourse) within a community sample. Further, we wanted to test whether the prevalence rates of sibling incest depend on sibling type (e.g., full versus step-siblings). To address our primary aim, we examined the following research questions: (1) Does sibling relationship type (full, half, or step-siblings) or participant sex (i.e., male or female) influence the likelihood of sibling incest? (2) Are hypothesized risk factors—including cues of relatedness (close proximity, maternal-neonatal

association, physical resemblance), family dysfunction (parent-child abuse, antisocial parents, nudity in the home), antisocial tendencies (antisocial behaviour, impulsivity, social popularity in childhood), atypical sexuality (sexual interest in children, hypersexuality), and emotions (incest disgust, sexual interest in sibling)—related to engaging in sibling incest (regardless of whether the participant or their sibling had instigated the sexual contact)? We also examined whether there were any differences between sibling incest that was coercive (i.e., person did not consent, use of force or threat, age gap > 5 years) versus incest that did not meet coercion criteria.

## Methods

### Study design and participants

This research consisted of a large cross-cultural online survey of 1,863 individuals ($N_{north\ America}$ = 909, $N_{Germany}$ = 691). A power analysis suggested that at least 1,000 participants were required to have sufficient power to detect unique correlates of sibling incest, however, more participants were recruited to have sufficient statistical power to conduct subgroup analyses. Recruitment took place online–the survey was posted on social media, research-designated websites (e.g., Psychological Studies on the Net), and on classified sites (e.g., Craigslist). Participants who came across the recruitment notice (e.g., on social media) were able to self-enrol in the study. Participants had to be 18 years of age or older, proficient in English or German, and had to have at least one opposite-sex sibling. Participants who did not meet these criteria were not able to complete the survey.

### Procedures

An online survey collected self-report information on all measures of interest (see S1 File for operational definitions and measure list). S1 File contains information about the internal validity of study measures. For this study, sibling incest was defined as any sexual behaviour between siblings (passionate kissing, touching sexual organs, masturbation, receiving or giving oral sex, vaginal or anal intercourse). The main outcome variable was whether sibling incest had occurred, regardless of whether the participant or their sibling had instigated the sexual contact. Participants were recruited predominantly from the United States, Canada, and Germany, using online platforms (S1 Table). Recruitment began in January 2017 and ended in December 2018.

The survey was created using Checkbox and was completed online. The survey was completely anonymous, and no third party had access to the data. The data were collected and stored on a server that was owned by, and located within, the [blinded for peer review]. After completing the survey, participants were directed to a second survey where they could enter their email to be entered for a chance to win 1 of 200 CA$25 Amazon gift cards (S2 File). Ethics was received by the [blinded for peer review]. All participants provided informed consent by clicking "agree" on the consent form presented online. Participants who provided informed consent were then directed to the survey.

### Statistical analysis

Three logistic regression models were examined to assess whether correlates were independently related to three outcomes: (1) whether any sibling incest had occurred (regardless of whether the participant or their sibling had instigated the incest), (2) whether the participant had initiated any coercive sexual contact, and (3) whether the participant had initiated any non-coercive sexual contact. Each outcome was dichotomous, such that participants who had

engaged in any of the outcomes (any sibling incest, instigating coercive or non-coercive incest) were compared to the participants who had never engaged in any sibling incest (i.e., sibling incest was never initiated by the participant or their sibling). Logistic regression yields an adjusted odds ratio (AOR) that reflects the relationship between a correlate and the outcome when controlling for the other correlates in the model. Based on the work of Mann et al. [20] AOR < 0.76 and AOR > 1.31 (equivalent to a Cohen's $d$ of -0.15 and 0.15, respectively) were considered meaningfully large. In other words, AOR < 0.76 indicate that a higher score on a correlate is associated with a meaningful decrease in the chance of an outcome occurring. Conversely, AOR > 1.31 suggest that a higher score on a correlate is associated with a meaningfully higher chance of an outcome occurring.

Based on the results of the logistic regressions, we conducted an exploratory mediation analysis. Mediation allows researchers to detect whether the effect of an independent variable on a dependent variable occurs because of the relationship between the independent variable and a mediator variable. The relationship between the independent variable and the dependent, through the mediator, is called the indirect effect. Indirect effects that reach statistical significance thresholds (i.e., $p < .05$) indicate that the independent variable is related to the dependent variable, in part, because of the mediating variable.

### Role of the funding source

The funder of the study had no role in study design, data collection, data analysis, data interpretation, or writing of the report.

## Results

Participants were 1,863 adults with at least one opposite-sex sibling who completed an online survey that assessed theoretically- or clinically-derived factors that could explain sibling incest. Demographic information is included in Table 1. Participants were predominantly female (66.4%; 1,237/1,863) and the plurality were North American (48.8%; 909/1,863). Participants were, on average, 27 years old (range: 18–65), and had two siblings (range: 1–13).

### Rates and characteristics of sibling incest

We found that rates of sexual contact between siblings were similar across Germany (12.1%, 93/773) and North America (13.5%; 146/1,085). Younger children were more likely to report engaging in sexual behaviours with a sibling than were older children (S1 Table). The mean age of onset was 10.5 (SD = 4.0; $n = 1,635$). Rates increased as genetic relatedness decreased (Fig 1), and males were more likely to report engaging in sexual behaviour with a sibling than females (males: 15.8%, 98/619; females: 11.5%, 141/1,230).

The main reasons given for having been involved in sexual behaviour between siblings were "curiosity" and "game," however, the reasons differed depending on the age of onset and sex of participants. Using age as a crude proxy for pubertal status, we split sibling incest cases between those occurring before the participant was 12 and those occurring when the participant was 12 years of age and older. Games was more likely to be the reported reason for sibling incest occurring before the age of 12, whereas desire and romance were more likely to be the reported reason for sibling incest occurring at the age of 12 or older; consummatory behaviours (i.e., vaginal intercourse) occurred more frequently when the contact occurred at the age of 12 and older (38.5%) than when it occurred prior to the age of 12 (29.8%; S1 Table, S1 Fig). Males were more likely to report desire and romance as reasons for engaging in sexual behaviour with a sibling compared to females (S2 Table, S2 Fig). In contrast, females were more likely to indicate curiosity as the reason for engaging in sexual behaviour with a sibling.

**Table 1. Demographics.**

| | | Frequency | |
| --- | --- | --- | --- |
| | | (*N* = 1,863) | |
| | | % | *n* |
| Gender | | | |
| | Woman | 65.1 | 1,212 |
| | Man | 33.9 | 632 |
| | Neither man nor woman | 0.4 | 8 |
| | Prefer not to respond | 0.6 | 11 |
| Sex | | | |
| | Female | 66.4 | 1,237 |
| | Male | 33.5 | 624 |
| | Neither male nor female | 0.1 | 2 |
| Region/Country | | | |
| | North America | 48.8 | 909 |
| | Germany | 37.1 | 691 |
| | Other countries[a] | 14.1 | 261 |
| | Prefer not to respond | 0.1 | 2 |
| Recruitment location | | | |
| | Research designated sites[b] | 39.9 | 744 |
| | Social media | 37.1 | 692 |
| | Classified sites[c] | 21.2 | 395 |
| | Prefer not to respond | 1.7 | 32 |
| Highest level of education achieved | | | |
| | Grade 8 or lower | 0.6 | 11 |
| | Some secondary school | 3.1 | 57 |
| | Secondary school diploma | 7.9 | 147 |
| | Some post-secondary school | 20.8 | 388 |
| | College diploma | 10.9 | 203 |
| | Apprenticeship or Trades certificate | 16.2 | 302 |
| | Bachelors degree | 27.1 | 505 |
| | Masters degree | 11.1 | 207 |
| | Doctorate | 1.9 | 36 |
| | Prefer not to respond | 0.4 | 7 |
| | | Mean (SD) | Range |
| Age | | 26.8 (8.4) | 18–65 |
| Total number of siblings[d] | | 2.3 (1.6) | 1–13 |
| Number of opposite-sex siblings[d] | | 1.7 (1.1) | 1–12 |

*Note*. [a]The most common were Austria (*n* = 79; 4.2%), the United Kingdom (*n* = 57; 3.1%), and Australia (*n* = 21; 1.1%). [b]Examples include r/SampleSize, Psychological Studies on the Net, and [blinded for peer review]'s research website. [c]Examples include Kijiji and Craigslist. [d]These variables had outliers; these values reflect the data after the outliers were reduced.

We also examined whether the age gap between a participant and their sibling influenced the prevalence and characteristics of sexual behaviour between siblings (S3 Table). Participants who had been involved in sexual behaviour with a sibling that was at least 5 years older reported being forced (27.5%; 11/40) more frequently than those with an age gap of 4 or fewer years (12.6%; 21/167, respectively; S3 Table, S3 Fig).

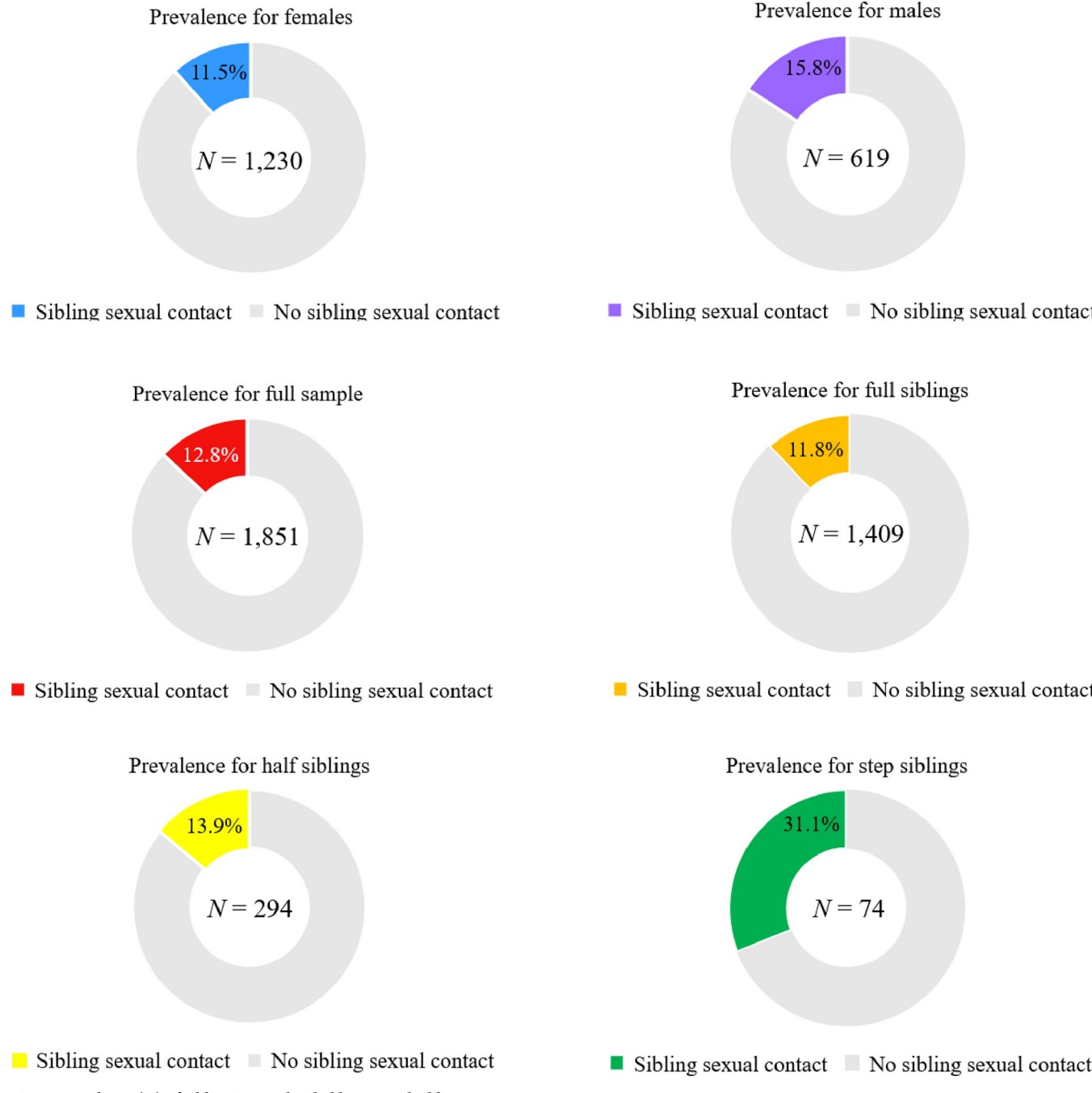

**Fig 1. Prevalence (%) of sibling incest divided by sex and sibling type.**

## Correlates of sibling incest

Most study variables were significant correlates of experiencing sibling incest at the bivariate level, but the only unique predictors (after controlling for other variables in the model) were positive family attitudes towards nudity, sexual contact with a parent, having a sexual interest in children, more atypical sexual behaviour in childhood (e.g., putting hand in underwear whilst in public), and less disgust towards sibling incest (see Table 2).

**Table 2.  Correlates associated with sibling incest.**

| | Any sibling incest | | | |
|---|---|---|---|---|
| | **AOR** | *p* | **AOR** | *p* |
| | **[95% CI]** | | **[95% CI]** | |
| Biological sex (male; *n* = 1,849) | **1.45 [1.10, 1.92]** | **.008** | 0.81 [0.51, 1.28] | .358 |
| **Cues of Relatedness** | | | | |
| Close proximity (*n* = 1,828) | **0.83 [0.74, 0.94]** | **.004** | 1.03 [0.84, 1.26] | .803 |
| Maternal-neonatal association (*n* = 1,768) | 0.96 [0.87, 1.06] | .453 | 0.97 [0.84, 1.12] | .671 |
| Physical resemblance (*n* = 1,846) | **0.92 [0.85, 0.99]** | **.034** | 0.93 [0.83, 1.05] | .234 |
| **Family Dysfunction** | | | | |
| Sexual abuse by a parent (yes; *n* = 1,788) | **9.20 [6.07, 13.93]** | **< .001** | **3.67 [1.87, 7.20]** | **< .001** |
| Childhood neglect (*n* = 1,837) | **2.23 [1.85, 2.69]** | **< .001** | 1.15 [0.75, 1.76] | .516 |
| Antisocial parents (*n* = 1,516) | **1.75 [1.39, 2.19]** | **< .001** | 0.91 [0.60, 1.37] | .657 |
| Positive family attitudes toward nudity (*n* = 1,829) | **1.43 [1.29, 1.60]** | **< .001** | **1.28 [1.09, 1.51]** | **.003** |
| **Antisocial Tendencies** | | | | |
| Childhood antisociality (*n* = 1,842) | **1.17 [1.09, 1.26]** | **< .001** | 0.96 [0.85, 1.09] | .524 |
| Childhood impulsivity (*n* = 1,842) | **1.49 [1.26, 1.76]** | **< .001** | 1.19 [0.92, 1.55] | .183 |
| Childhood popularity (*n* = 1,845) | 1.00 [0.87, 1.14] | .947 | 0.83 [0.69, 1.01] | .067 |
| **Atypical Sexuality** | | | | |
| Sexual interest in children (*n* = 1,839) | **1.26 [1.19, 1.34]** | **< .001** | 1.13 [1.03, 1.24] | .012 |
| Atypical childhood sexual behaviours (*n* = 1,754) | **1.18 [1.15, 1.21]** | **< .001** | **1.11 [1.07, 1.16]** | **< .001** |
| Hypersexuality (*n* = 1,836) | **1.29 [1.19, 1.40]** | **< .001** | 1.09 [0.97, 1.24] | .147 |
| **Proximal Factors** | | | | |
| Disgust toward sibling incest (*n* = 1,838) | **0.60 [0.55, 0.66]** | **< .001** | **0.75 [0.65, 0.87]** | **< .001** |
| Sexual interest in sibling (*n* = 1,828) | **1.48 [1.38, 1.58]** | **< .001** | 1.13 [0.99, 1.29] | .075 |

*Note*. AOR = adjusted odds ratios; odds ratios adjusted for other variables in the model. $R^2$ Model$_{Any sexual behaviour}$ = .29 (*N* for AOR model = 1,311; number of incidents = 239). Place of data collection did not add to the multivariate model (AOR = 0.90, 95% CI [0.56, 1.44]). Odds ratios for maternal-neonatal association for older sibling sets (OR = 1.12, 95% CI = [0.95, 1.32], *n* = 792; AOR = 1.08, 95% CI [0.86, 1.35], *n* = 586). Bolded value reached statistical significance at p < .05.

Given that proximity was not a unique predictor of sibling incest, and given that the relationship between sexual interest in a sibling and sibling incest attenuated when controlling for other variables (Table 2), we tested whether sexual interest mediated the relationship between proximity and sibling incest. We found a small, significant, indirect effect, suggesting that sexual interest in a sibling mediated the relationship between constant proximity and sibling incest (Fig 2, S4 Table).

Further, place of data collection (North America or Germany) did not contribute to the model (see Table 2), and therefore we concluded that the effects of the examined correlates on sibling incest were not different across English- and German-speaking participants.

We defined coercive sibling incest as sexual contact between siblings that was described as non-consensual by the participant (i.e., the participant and/or their sibling had not consented), that involved an age difference of 5 years or greater, and/or where physical force was used [14]. Half of all incidents were classified as coercive sibling incest (i.e., either the participant or their sibling had not consented; 49%). More female participants reported being coerced into engaging in sexual behaviours with a sibling (34%) than male participants (11%; S5 Table). Only a minority of incidents of coercive sibling incest led to an arrest (12.3%; 14/114). Positive family attitudes towards nudity was a unique correlate for the perpetration of non-coercive sibling

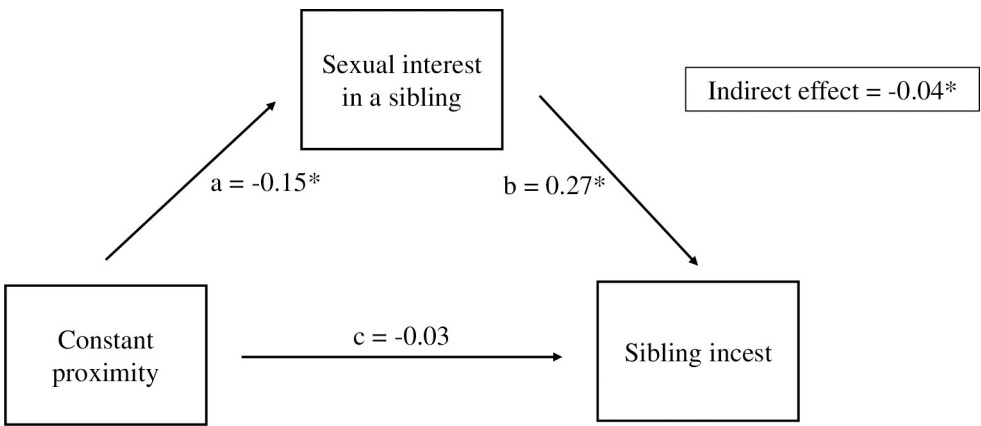

**Fig 2. Mediating effect of sexual interest in a sibling.** Asterisks denote significant effects, $p < .001$. $N = 1,807$. Values represent standardized coefficients. By itself, constant proximity was not significantly related to whether sibling incest had occurred. Instead, constant proximity was related to a decreased chance that sibling incest had occurred, through its relationship with sexual interest in a sibling.

incest whereas impulsivity and sexual interest in children were unique correlates for the perpetration of coercive sibling incest (see Fig 3, S6 Table).

## Discussion

Our data provides evidence that sexual behaviour between siblings is not rare, especially in families with lower relatedness between siblings (e.g., step or half-siblings), when there are other problematic sexual behaviours occurring in childhood, when nudity is accepted in the home, and when there is less disgust toward the idea of sexual contact with a sibling. In this survey of 1,863 English- or German-reading adults with at least one opposite-sex sibling, we found that approximately 1 in 10 had engaged in sibling incest, and half of these incidents

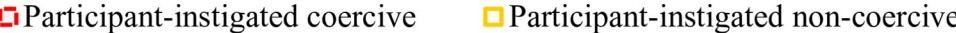

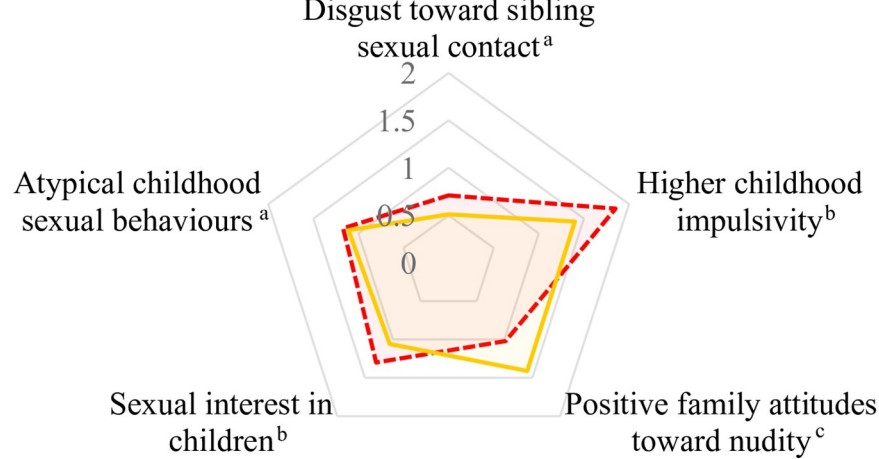

**Fig 3. Comparison of correlates (adjusted odds ratio) across sibling incest type.** $n_{Coercive} = 1,190$; $n_{Non-coercive} = 1,213$. Only correlates that had a significant AOR value for either group (i.e., participant coercive or participant non-coercive) were included in the figure. [a]Significant for both outcomes at $p < .05$. [b]Significant for participant-instigated coercive at $p < .05$. [c]Significant for participant-instigated non-coercive at $p < .05$.

could be classified as coercive (i.e., one sibling did not consent, age gap was greater than 5 years, and/or force was used). There were notable differences between those who had experienced sexual behaviour with a sibling prior to the age of 12 (reported more as play) versus at the age of 12 or older (meeting a sexual desire). There were also notable differences between male and female participants, with males being more likely to report desire and romance as reasons for sibling incest, and females being coerced into sibling incest more.

A large age gap between siblings is often used as an indicator of problematic sexual behaviour because of developmental differences between the siblings in terms of cognition, emotional and social development, and autonomy [14]. Our findings are consistent with considering large age gap as an indicator because the use of force was more common as the age gap between siblings widened. Additionally, perpetrating non-coercive and coercive sibling incest had unique and overlapping correlates suggesting they are different phenomena.

Rates of consummatory sibling incest vary in the literature. In clinical or legal settings the frequency of vaginal intercourse has been reported to occur in as few as one in ten cases [21, 22] to as high as seven [23] or nine in ten [24]. Although lower in community and university student populations, around one-third of sibling incest seems to be consummatory (21.4% [25]; 36.6% [26]; 36.9% [27]; 60.7% [28]), which is in line with the findings in this study. Consummatory sibling incest can lead to reproduction and thus risks inbreeding depression, where any resulting offspring are higher in morbidity and mortality due to the combination of deleterious recessive alleles [6, 29]. Inbreeding depression is the putative selection pressure behind incest avoidance, and thus consummatory behavior should be the rarest form of sibling incest.

In line with past research, we found that cohabitation and physical resemblance had significant bivariate relationships with sibling incest in our sample of 1,863 participants (all of whom had at least one opposite-sex sibling) [7, 8, 10, 25, 28, 30]. However, more cohabitation, seeing more parental associations between siblings and mother, and physical resemblance did not significantly reduce sexual behaviour between siblings, after controlling for other (more proximal) correlates. Although differences were in the expected direction, proximity, maternal-neonatal association, and resemblance were not unique predictors of sexual behaviour between siblings. It is possible that proximal factors mediate the relationship between cues of relatedness and sibling incest. Sexual interest in one's sibling was one of the strongest bivariate correlates of sibling incest, but this relationship attenuated when controlling for other factors. Moreover, proximity and sexual interest in a sibling were moderately negatively related (S7 Table). Indeed, we found that sexual interest in a sibling mediated the relationship between proximity and sibling incest, suggesting that some of the effect of cohabitation on sibling incest is because of sexual interest in one's sibling.

Past research has primarily focused on bivariate relationships between risk factors and sibling sexual contact [8, 30], but the current study reveals that when considered together, other factors (e.g., family dysfunction) matter more.

While disgust was related with sibling incest, and proximity and disgust had a small correlation, disgust was not related to maternal-neonatal association, nor physical resemblance, which is not consistent with past research [8, 9, 30]. Different measures assessed each of these constructs in all four studies; to the authors' knowledge, no validated measures of cues of relatedness exist. It is possible that slight variations between items assessing cues of relatedness across studies have contributed to some inconsistency in findings in the literature. Future research should endeavour to validate measures of relatedness, as this could be a major limitation within the field. However, it is also possible that disgust is more driven by taboos than sexual interest in a sibling, and so future research should measure and control for incest taboos (e.g., personal, cultural, or religious beliefs).

Reporting other problematic sexual behaviour in childhood increased the likelihood of having engaged in sibling incest, suggesting that professionals working with children with problematic sexual behaviour should assess for co-occurring sibling incest. Family acceptance of nudity, sexual abuse by parents, and lower disgust to sibling incest could be promising markers of sibling incest. The current findings also highlight the importance of sexual education, especially in blended households, where age gaps and lower genetic relatedness were associated with more sexual behaviour between siblings.

Sibling intrafamilial child sexual abuse is amongst the most underreported forms of sexual abuse [1, 4]. As such, survey designs are essential for research on sibling incest given official data are underreported (or not reported in the case of consensual sexual behaviour between similar-aged siblings). That said, in the present study participants reported whether they or their sibling had consented to the sibling sexual contact. As such, endorsement of non-consent may be biased because the participant (willfully or unintentionally) mischaracterized whether their sibling had consented. Future research should replicate these results in non-WEIRD countries (Westernized, Industrialized, Educated, Rich, Democratic [31]) and testing other predictors suggested by anthropological and psychological research on incest. The present study examined the correlates of sexual behaviour between siblings (i.e., any sexual behaviour between a participant and their sibling, regardless of who instigated the contact). We also examined the correlates of coercive and non-coercive sibling incest perpetrated by the participant, though the sample sizes (participant-instigated coercive $n = 49$; participant-instigated non-coercive sibling incest $n = 86$) preclude strong conclusions. Future research should endeavour to examine the correlates of instigating sibling incest, specifically.

## Conclusion

Drawing from an online survey of 1,863 siblings, we found that 1 in 10 siblings engaged in sexual behaviour with another sibling. This rate is higher for blended families. We found a comorbidity between higher rates of childhood sexual behaviours and sibling incest. As such, professionals who assess and treat children with problematic sexual behaviour and children who experienced sexual abuse by a parent should investigate for inappropriate interaction between siblings as they often co-occur. Positive familial attitude toward nudity is associated with a greater chance of sibling incest. Participants who were more impulsive and who had a sexual interest in children were more likely to have instigated coercive sibling incest against their sibling, while children who grew up in families that espoused positive attitudes about nudity were more likely to instigate non-coercive sexual contact with a sibling. Given that the present study found sibling incest to be more common in blended families (i.e., between non-full siblings), our findings highlight the importance of sexual education within blended households.

## Supporting information

**S1 File. Detailed measures.**
(PDF)

**S2 File. Detailed procedure.**
(PDF)

**S3 File. Questionnaire on inclusivity in global research.**
(DOCX)

**S4 File. Inclusivity in global research.**
(DOCX)

**S1 Table. Characteristics of sibling incest by age contact began.**
(PDF)

**S2 Table. Characteristics of sibling incest by participant sex.** Matching superscripts within rows indicate that the values are not significantly different at $p < .05$. Superscripts that do not match within rows indicate the values are different at $p < .05$.
(PDF)

**S3 Table. Characteristics of sibling incest by age gap between siblings.** Matching superscripts within rows indicate that the values are not significantly different at $p < .05$. Superscripts that do not match within rows indicate the values are different at $p < .05$. [c]Participants reported whether they or their sibling had ever consented to sibling incest. If there was any evidence of non-consent (i.e., participants reported they or their sibling had not consented), 'yes' was scored for this variable.
(PDF)

**S4 Table. Mediating effect of sexual interest in a sibling.** Values represent standardized coefficients.
(PDF)

**S5 Table. Coercive sibling incest by participant sex.** Coercion occurred if, in an instance of sibling incest, either sibling did not consent (i.e., the participant reported that they or their sibling had not consented), there was an age gap of more than 5 years between the siblings, or force was used. Matching superscripts within rows indicate that the values are not significantly different at $p < .05$. Superscripts that do not match within rows indicate the values are different at $p < .05$.
(PDF)

**S6 Table. Correlates associated with sibling incest by coerciveness.** AOR = adjusted odds ratios; odds ratios adjusted for other variables in the model. Sibling incest was defined as coercive if there was no consent, if force or threat were used to achieve the contact, or if there was an age difference of 5 years or more between siblings. Non-coercive sibling incest was consensual, did not involve force or threats, and occurred between siblings with less than a 5-year age gap. $R^2$ Model$_{\text{Coercive}}$ = .41 ($n$ = 1,190/1,863; number of incidents = 49); $R^2$ Model$_{\text{Non-coercive}}$ = .30 ($n$ = 1,213/1,863; number of incidents = 86). Sample sizes for coercive ORs ranged from 1,376 to 1,659 (mdn = 1,646) and for non-coercive ranged from 1,401 to 1,696 (mdn = 1,683).
(PDF)

**S7 Table. Pearson correlation between cues of relatedness and proximal factors.** [a] $p < .001$, [b] $p < .01$. $n_{\text{Proximity}}$ between 1,818 and 1,828; $n_{\text{Maternal-neonatal}}$ between 1,764 and 1,772; $n_{\text{Resemblance}}$ between 1,838 and 1,848.
(PDF)

**S1 Fig. Reasons for sibling incest by age sexual contact began.** Asterisks denote significant group differences, p < .05, all n between 123 and 126.
(PDF)

**S2 Fig. Reasons for sibling incest by sex.** Asterisks denote significant group differences, p < .05, all n between 216 and 222.
(PDF)

**S3 Fig. Proportion of coercive characteristics of sibling incest by age gap.** Asterisks denote significant group differences, $p < .05$, all $n$ between 207 and 227.
(PDF)

## Author Contributions

**Conceptualization:** Kelly M. Babchishin, Rainer Banse, Lisa Huppertz.

**Formal analysis:** Kelly M. Babchishin, Emma J. Holmes.

**Funding acquisition:** Kelly M. Babchishin, Michael C. Seto.

**Methodology:** Kelly M. Babchishin, Rainer Banse, Lisa Huppertz, Michael C. Seto.

**Resources:** Michael C. Seto.

**Supervision:** Rainer Banse.

**Writing – original draft:** Kelly M. Babchishin, Michael C. Seto.

**Writing – review & editing:** Kelly M. Babchishin, Emma J. Holmes, Rainer Banse, Lisa Huppertz, Michael C. Seto.

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
