## [Decision Letter · Decision Letter 0]

13 Aug 2024

PONE-D-24-20078Characteristics and Risk Factors for Sibling IncestPLOS ONE

Dear Dr. Banse,

Thank you for submitting your manuscript to PLOS ONE. After careful consideration, we feel that it has merit but does not fully meet PLOS ONE’s publication criteria as it currently stands. Therefore, we invite you to submit a revised version of the manuscript that addresses the points raised during the review process.

We look forward to receiving your revised manuscript.

Kind regards,

Tijani Idris Ahmad Oseni, FMCFM

Academic Editor

PLOS ONE

3. Please amend your authorship list in your manuscript file to include author Lisa Huppertz.

Additional Editor Comments:

The subject matter is of relevance and the background was well written.

however, the aim of the study was not well spelt out in the manuscript.

Also, the methods section is greatly flawed and need revision. for instance,

how was the sample size determined?

How were participants recruited?

What determined eligibility to receive the online survey?

What were the inclusion and exclusion criteria?

How was the study instrument developed and validated?

How reliable is the instrument for data collection?

The Table one in the study design should actually be in the results section.

Reviewers' comments:

Reviewer's Responses to Questions

**Comments to the Author**

1. Is the manuscript technically sound, and do the data support the conclusions?

Reviewer #1: Yes

Reviewer #2: Yes

2. Has the statistical analysis been performed appropriately and rigorously? 

Reviewer #1: Yes

Reviewer #2: Yes

3. Have the authors made all data underlying the findings in their manuscript fully available?

Reviewer #1: Yes

Reviewer #2: Yes

4. Is the manuscript presented in an intelligible fashion and written in standard English?

Reviewer #1: Yes

Reviewer #2: Yes

5. Review Comments to the Author

Reviewer #1: It is an interesting research. It brought a lot of factors to bear under the correlates. The result answered the highlighted researvh questions. I understand that there a dearth of research in this field however I feel like under the discussion, findings from similar research should be compared with the findings from this study. Possible reasons for the observed difference(s) can then be brought out e.g methodological differences, sociocultural reasons etc.These reasons can spur future research in this field. Some part of the discussion tried to follow this pattern however most part did not.

Reviewer #2: Passionate kissing among sibling may not be regarded as sexual relations. you need to define it more and give backing

Reference number 6. no date when internet was accessed.

Reference 19, please include when the internet was accessed

Reference number 20. include the month as you did with others references.

In second survey, incentive of gift may create bias in the person taking the survey.

6. PLOS authors have the option to publish the peer review history of their article (what does this mean?). If published, this will include your full peer review and any attached files.

Reviewer #1: No

Reviewer #2: **Yes: **Dr Kumbet John Sonny

---

## [Author Response · Author response to Decision Letter 0]

10 Oct 2024

Dear Dr. Oseni,

We are pleased to receive the chance to revise and resubmit our manuscript "Characteristics and risk factors for sibling incest" (Manuscript No. PONE-D-24-20078) to PLOS ONE. 

We have addressed all comments received in the uploaded revised version of the manuscript and summarized changes that pertain to each in the list below (original comments in bold).

Thank you again for the helpful comments. We feel that the revised manuscript is much stronger because of these comments. 

The authors.

We have updated the manuscript to meet the style requirements.

2. Please include a complete copy of PLOS’ questionnaire on inclusivity in global research in your revised manuscript. Our policy for research in this area aims to improve transparency in the reporting of research performed outside of researchers’ own country or community. The policy applies to researchers who have travelled to a different country to conduct research, research with Indigenous populations or their lands, and research on cultural artefacts. The questionnaire can also be requested at the journal’s discretion for any other submissions, even if these conditions are not met. Please find more information on the policy and a link to download a blank copy of the questionnaire here: https://journals.plos.org/plosone/s/best-practices-in-research-reporting . Please upload a completed version of your questionnaire as Supporting Information when you resubmit your manuscript.

We have added the questionnaire on inclusivity in global research to the supporting materials.

3. Please amend your authorship list in your manuscript file to include author Lisa Huppertz.

Lisa Huppertz was added to the authorship list.

Additional Editor Comments:

The subject matter is of relevance and the background was well written.

however, the aim of the study was not well spelt out in the manuscript.

Thank you. We have outlined our study aims in lines 89-93.

Also, the methods section is greatly flawed and need revision. for instance,

how was the sample size determined?

We have revised the manuscript so that lines 108-110 highlight how sample size was determined.

How were participants recruited?

What determined eligibility to receive the online survey?

What were the inclusion and exclusion criteria?

Lines 111-116 now outline how recruitment took place, how participants could enrol in the study, as well as our inclusion criteria.

How was the study instrument developed and validated? 

How reliable is the instrument for data collection?

S1 File includes information about study measures and cites any scales that were included. S1 File also includes a measure of internal consistency (Cronbach’s alpha) for any scales that were presented to participants.

The Table one in the study design should actually be in the results section.

Table 1 has been moved to the Results section.

Review Comments to the Author

Reviewer #1: It is an interesting research. It brought a lot of factors to bear under the correlates. The result answered the highlighted researvh questions. I understand that there a dearth of research in this field however I feel like under the discussion, findings from similar research should be compared with the findings from this study. Possible reasons for the observed difference(s) can then be brought out e.g methodological differences, sociocultural reasons etc.These reasons can spur future research in this field. Some part of the discussion tried to follow this pattern however most part did not.

Thank you. We have revised such that all sections of the discussion indicate whether our results are in line with past research or not. Lines 273-287 outline the major findings from our research which did not overlap with past research and we have provided potential reasons for this (e.g., that instead of being directly related to incest avoidance, proximal factors mediate the relationship between cues of relatedness and sibling incest). 

Reviewer #2: Passionate kissing among sibling may not be regarded as sexual relations. you need to define it more and give backing

Passionate kissing (e.g., open-mouthed kissing involving tongue contact) is a non-normative childhood sexual behaviour that is included in Friedrich et al’s Child Sexual Behavior Inventory. We have included this information in lines 72-75 of the manuscript.

Reference number 6. no date when internet was accessed.

Reference 19, please include when the internet was accessed

The date these resources were accessed was added.

Reference number 20. include the month as you did with others references.

The month was included in this reference.

In second survey, incentive of gift may create bias in the person taking the survey.

While incentives could bias who chose to enrol in the study, we mitigated this by offering a relatively low incentive. Further, incentivising survey participants through small gifts (i.e., gift cards) is a common technique used to recruit sample sizes that are sufficiently large to conduct multivariate regression.

---

## [Decision Letter · Decision Letter 1]

14 Nov 2024

Characteristics and risk factors for sibling incest

PONE-D-24-20078R1

Dear Dr. Rainer Banse,

We’re pleased to inform you that your manuscript has been judged scientifically suitable for publication and will be formally accepted for publication once it meets all outstanding technical requirements.

Kind regards,

Tijani Idris Ahmad Oseni, FMCFM

Academic Editor

PLOS ONE

Additional Editor Comments (optional):

Reviewers' comments:

Reviewer's Responses to Questions

**Comments to the Author**

1. If the authors have adequately addressed your comments raised in a previous round of review and you feel that this manuscript is now acceptable for publication, you may indicate that here to bypass the “Comments to the Author” section, enter your conflict of interest statement in the “Confidential to Editor” section, and submit your "Accept" recommendation.

Reviewer #1: All comments have been addressed

2. Is the manuscript technically sound, and do the data support the conclusions?

Reviewer #1: Yes

3. Has the statistical analysis been performed appropriately and rigorously? 

Reviewer #1: Yes

4. Have the authors made all data underlying the findings in their manuscript fully available?

Reviewer #1: Yes

5. Is the manuscript presented in an intelligible fashion and written in standard English?

Reviewer #1: Yes

6. Review Comments to the Author

Reviewer #1: The concerns have been addressed and the ethical approval has been stated. The identities of the respondents cannot be traced which is a major ethical consideration in this article. The article can be published

7. PLOS authors have the option to publish the peer review history of their article (what does this mean?). If published, this will include your full peer review and any attached files.

Reviewer #1: **Yes: **Odunaye-Badmus Sekinat Oloruntosin

---

## [Editor Report · Acceptance letter]

20 Nov 2024

PONE-D-24-20078R1 

PLOS ONE

Dear Dr. Banse, 

I'm pleased to inform you that your manuscript has been deemed suitable for publication in PLOS ONE. Congratulations! Your manuscript is now being handed over to our production team.

Kind regards, 

on behalf of

Dr. Tijani Idris Ahmad Oseni 

Academic Editor

PLOS ONE